# Degradation Product of Sea Cucumber Polysaccharide by Dielectric Barrier Discharge Enhanced the Migration of Macrophage In Vitro

**DOI:** 10.3390/foods12224079

**Published:** 2023-11-10

**Authors:** Shiwen Cheng, Han Cai, Meng Yi, Liang Dong, Jingfeng Yang

**Affiliations:** School of Food Science and Technology, Dalian Municipality Engineering Laboratory for Shellfish Polysaccharide, National Engineering Research Center of Seafood, Dalian Polytechnic University, Dalian 116034, China; 17866589745@163.com (S.C.); 13941001677@163.com (H.C.); yimeng2315a@163.com (M.Y.); dongzhongxiang@126.com (L.D.)

**Keywords:** dielectric barrier discharge, polysaccharide, degradation, RAW264.7 cell, migration

## Abstract

This study investigated the effect of dielectric barrier discharge (DBD) on sea cucumber polysaccharide (SP-2) and evaluated its anti-inflammatory properties. The SP-2 was depolymerized by applying an input voltage of 60~90 V for 3~9 min. The features of the products were examined using high-performance gel permeation chromatography, HPLC-PAD-MS, and the Fourier transform infrared (FTIR) spectrum. The anti-inflammatory properties of the product were investigated by measuring nitric oxide (NO) release, ROS accumulation, and cell migration using RAW264.7 cells (LPS-induced or not-induced). The results showed SP-2 depolymerized into homogeneous and controllable-size oligosaccharide products. The depolymerized ratio can reach 80%. The results of the measurement of reducing sugars indicate that SP-2 was cleaved from within the sugar chain. The SP-2 was deduced to have a monosaccharide sequence of GlcN-Man-Man-Man-Man-Man based on the digested fragment information. The depolymerization product restrained the release of NO and the accumulation of ROS. By testing the RAW264.7 cell scratch assay, it was found that it enhances the migration of immune cells. DBD degradation of SP-2 leads to homogeneous and controllable-size oligosaccharide products, and this technique can be used for polysaccharide structure analysis. The depolymerized product of SP-2 has an anti-inflammatory capability in vitro.

## 1. Introduction

The analysis of polysaccharides’ primary structure involves complex information that can be used to reconstruct molecular structure. The accuracy of the structure analysis of the polysaccharide is inhibited by molecules that are too large. The low-molecular-weight derivative can minimize the research limitations [1]. Thus, a precise depolymerization technique is crucial for analyzing the primary structure of polysaccharides. The depolymerization products should be small in size to facilitate structure analysis using diverse spectroscopy techniques [2].

The degradation product of sea cucumber polysaccharides provides stronger bioactivity and a facility for absorption, getting through the intestinal mucosa barrier. Until now, several polysaccharide degradation methods have been established. Standard degradation methods for polysaccharide structure analysis include partial acid hydrolysis and enzymatic hydrolysis methods [3,4]. Other degradation methods that have been developed cannot be employed in the study of structural analysis, such as Cobalt-60 (^60^Co) [5], hermetical microwave degradation, acid hydrolysis, and free-radical depolymerization [6,7]. However, the methods of partial acid hydrolysis and enzymatic hydrolysis are unable to cleave the polysaccharide at specific sites and therefore cannot meet the current demand.

A new method using dielectric barrier discharge (DBD) plasma was recently developed to degrade chemical substances, for example, N,N-diethyl-m-toluamide [8]. DBD plasma has the ability to create a more uniform discharge [9]. Hence, the DBD degradation method may encounter the need for polysaccharide structure analysis. 

Sea cucumber is a high-value seafood in East Asia. This species is considered both a medicine and a food source. The sea cucumber polysaccharide was reported to have a variety of biological effects, including morphological transformation and proliferation of astrocytes [10], modulating the gut microbiota [11], and anticoagulant activity [12]. However, the high molecular weight hampered the absorption of polysaccharides through intestinal membranes, resulting in greater activity [13]. Hence, the molecular-weight-controlled degradation of sea cucumber polysaccharide is not only essential for the food function proposed but also for the structure analysis.

Macrophages are essential for host defense and the maintenance of tissue homeostasis. When the body is injured or attacked by bacteria, large macrophages migrate to the injured site to perform their immune defense function [14,15]. In recent years, the RAW264.7 cell, as one of the main research models of macrophage cell lines, has been widely reported to participate in the first line of primary defense. Moreover, induction with lipopolysaccharide (LPS) can polarize RAW264.7 cells into an M1 pro-inflammatory phenotype, which causes the cells to release a large number of inflammatory factors, such as nitric oxide (NO) and reactive oxygen species (ROS). However, excessive accumulation of NO and ROS can further cause cytotoxic damage and chronic inflammatory diseases, seriously disrupting the balance of the immune system, such as arthritis and Alzheimer’s disease [16]. During this process, researchers have been widely concerned about searching for safe and effective bioactive substances to regulate the body’s immune homeostasis. 

In this study, a novel dielectric barrier discharge degradation method was adopted to depolymerize the sea cucumber polysaccharide and reconstruct the primary sequence of the polysaccharide. After that, the anti-inflammatory effect of the DBD-treated product on LPS-stimulated RAW 264.7 macrophages was analyzed in vitro. This research attempts to find an effective technique to depolymerize the polysaccharide to benefit polysaccharide structure analysis with the features of efficient, controllable, and homogeneous products. The anti-inflammatory capability of the DBD treatment product was also tested. This study introduces a new, effective method for depolymerizing polysaccharides without the use of added chemicals while maintaining a fast reaction rate. In addition, the study will also explore the anti-inflammatory effects and enhancement of macrophage migration by the degraded product. 

## 2. Materials and Methods

### 2.1. Materials and Reagents

Twelve sea cucumbers (each about 180 ± 10 g) were purchased from the local aquatic product market (Voucher number of 6,937,774, Dalian, 39°1′8″ N, 122°45′34″ E, China), and morphology confirmed that all the samples belonged to *Stichopus japonicus*. Then, all sea cucumbers were freeze-dried immediately after their arrival at the lab. *S. japonicus* was mashed and stored at 4 °C for further extraction. The trypsin and pepsin were purchased from Sanland Chemicals Co. (Shanghai, China). This study used only analytical-grade or above-grade chemicals. Standard monosaccharides, standard MW dextran, and MD-25 dialysis tubes (MWCO: 3 kDa) were purchased from Pharmacia Co. (Uppsala, Sweden), and Sepharose CL-6B was purchased from Amersham Co. (Uppsala, Sweden). RAW264.7 cells were purchased from Abiowell (Changsha, China), DMEM was purchased from Gibco (Grand Island, NY, USA), and FBS was purchased from Sigma (St Louis, MO, USA).

### 2.2. Isolation and Purification of Sea Cucumber Polysaccharide

The extraction method was operated as described by Wang with a small modification [17]. Thirty g of sea cucumber powder were added to a 2% sodium hydroxide solution at a concentration of 5%. The solution was then incubated at 60 °C for 3 h. This solution was then adjusted to pH 7 with 6 M hydrochloric acid and centrifuged at 4000× *g* for 15 min. The supernatant was adjusted to pH 2.3 and hydrolyzed by pepsin for 5 h at 50 °C. After cooling to room temperature, the solution was adjusted to pH 8.2 with 4 M sodium hydroxide and hydrolyzed by trypsin for another 5 h at 52 °C. Then the solution was heated to 90 °C for 10 min to deactivate the enzymes and adjusted to pH 7 with 6 M hydrochloric acid. After centrifuging at 4000× *g* for 15 min, the supernatant was collected, and a threefold volume of 95% ethanol (*v*/*v*) was added for polysaccharide precipitation. The incubated mixture was held for 12 h at 4 °C, then the precipitate was collected and washed twice with absolute ethanol. The precipitate was dissolved in deionized water at a concentration of 5% (*w*/*v*), and the solution was subsequently dialyzed for 3 days. The supernatant was freeze-dried to obtain the crude sea cucumber polysaccharide (SCP).

The hydrolyzed proteins in crude sea cucumber polysaccharide were removed using the Sevag method [18]. Then the glycogen component was removed, according to Yang [19]. Further, the sample was purified by gel filtration chromatography using a Sepharose CL-6B column (1.6 cm × 70 cm, Amersham Biosciences, Uppsala, Sweden), equilibrated with 0.15 M NaCl. Finally, the separation products were recovered, dialyzed, and lyophilized to obtain the purified sea cucumber polysaccharide fraction SP-2. 

### 2.3. Degradation by Dielectric Barrier Discharge (DBD)

A DBD instrument operated at atmospheric pressure was used for the degradation of SP-2. The experiments were carried out in a 500 W atmospheric pressure DBD instrument (CTP-2000K, Nanjing Suman Plasma Technology Co., Ltd., Nanjing, China) with a frequency of 1 to 100 kHz and a maximal voltage output of 30 kV. This experimental apparatus contains a plasma generator, a dielectric barrier plasma reactor, and a cylindrical quartz reaction chamber (12 mm in height and 6 cm in inner diameter). The dielectric barrier plasma reactor consists of top and bottom electrodes, with the cylindrical quartz reaction chamber in between. Plasma treatments were performed in the cylindrical quartz reaction chamber that surrounds the sample. The SP-2 depolymerization procedure by the DBD method was operated as described in our previous report [20]. SP-2 was dissolved in 5% (*w*/*v*) distilled water. One mL of the solution was transferred to the reaction chamber for DBD depolymerization. The study investigated the effects of treatment time (3–9 min), input voltage (60–90 V), and sample concentration (1–5 mg/mL) on depolymerization. Then the DMB (1,9-dimethylmethylene blue) method was used to detect the sulfate group [21].

### 2.4. HPGPC Analysis

High-performance gel permeation chromatography (HPGPC) was used to monitor the purification of SP-2 and the molecular weight (MW) change during SP-2 depolymerization. It includes a liquid chromatography system, Waters e2695 (Waters Co., Milford, MA, USA), equipped with a 7.8 × 300 mm TSK-gel G4000PWXL column (Tosoh Bioscience, Tokyo, Japan), and a 2414 Refractive Index Detector (Waters Co., Milford, MA, USA). Sample solutions (20 μL per run) were injected into the injector using distilled water as the mobile phase at a flow rate of 0.2 mL/min. The column was calibrated using dextran standards (1 × 10^3^ Da, 1.2 × 10^3^ Da, 5 × 10^3^ Da, 4.7 × 10^4^ Da, 5 × 10^4^ Da, and 6.1 × 10^4^ Da). 

### 2.5. FTIR Spectroscopic Analysis

The Fourier transform infrared (FTIR) spectra were recorded using a Specord 75 IR spectrometer (Carl Zeiss, Jena, Germany) with KBr pellets.

### 2.6. Analysis of Monosaccharide Composition

The polysaccharide samples were derivatized by PMP, according to Wang [2]. Then, the PMP derivatives were subjected to HPLC-PAD-MS for analysis. The HPLC-PAD-MSn analysis was performed on an LXQ linear ion trap mass spectrometer (Thermo Fisher Scientific, Basel, Switzerland) with a Silgreen ODS C18 (250 × 4.6 mm, 5 μm) column. The mobile phase was a mixture of 0.02 M ammonium acetate/acetonitrile (84:16, *v*/*v*). 

### 2.7. Measurement of Reducing End Formation in Polysaccharides Treated with DBD

The DBD-treated polysaccharide was tested for reducing end formation using a 3,5-dinitrosalicylic acid (DNS) method [22]. An aliquot of the polysaccharide solution treated with DBD was dissolved in distilled water and diluted 100-fold. One mL of these sample solutions (treatment times ranging from 0, 3, 5, and 7 min) were mixed with 1.5 mL of DNS reagent, respectively. The mixtures were subsequently incubated in a boiling water bath for 5 min. After being cooled down to room temperature, the solutions were diluted another 25 times with distilled water, and the absorbance of the solution was measured (Infinite M200, Tecan Infinite, Switzerland) at 520 nm. The blanks were prepared in the same way as the analyzed sample, replaced by distilled water.

### 2.8. RAW264.7 Cell Culture

RAW264.7 cells were routinely cultured in Dulbecco’s Modified Eagle Medium (DMEM) supplemented with 10% fetal bovine serum (FBS), 100 U/mL penicillin, and 100 µg/mL streptomycin. The cells were cultivated at 37 °C and 5% CO_2_ in a damp environment. The fresh medium was replaced every 2 days.

### 2.9. Cell Toxicity Assay

To prepare the depolymerization product for this assay, the same procedure described in Section 2.3 was followed. SP-2 was depolymerized by DBD with a power of 71.19 W and a maximal voltage output of 30 kV, and the product was designated as Oligo-SP-2. The effect of SP-2 or Oligo-SP-2 on the viability of RAW264.7 cells were determined by the MTT method [23]. RAW264.7 cells were seeded at a density of 1 × 10^5^/mL in 96-well plates (100 μL per well). Then the cells were co-cultured with SP-2 or Oligo-SP-2 for 24 h at concentrations of 1, 5, 10, 20, 25, and 50 μg/mL. After that, 10 μL of MTT solution (5 mg/mL) was added to each well and incubated for 4 h at 37 °C in the dark under 5% CO_2_. After the incubation, the solution was removed from the wells and replaced with 150 μL of dimethyl sulfoxide (DMSO) to dissolve the purple formazan crystals in the cells. The absorbance of each well was measured at 490 nm with a Microplate Reader (Infinite M200, TECAN, Männedorf, Switzerland).

### 2.10. Nitric Oxide (NO) Content Determination Assay

The content of NO in RAW264.7 cells after treatment with SP-2, Oligo-SP-2, and LPS was determined according to the NO kit [24]. Cells were seeded at a density of 1 × 10^5^ cells/well in the 96-well plates and then incubated for 24 h at 37 °C in a 5% CO_2_ incubator (6 replicates/group). After this incubation period, the culture medium was replaced with 100 µL of SCP, SP-2, Oligo-SP-2, and LPS solutions and incubated for 24 h. The sample solution contained LPS with a concentration of 1 μg/mL and SP-2 or Oligo-SP-2 with a concentration of 0, 1, 5, 10, 20, 25, and 50 μg/mL, respectively. The 0 μg/mL Oligo-SP-2 group was used as the control group, and 1 μg/mL LPS was the model group. The positive control group was 20 μg/mL diclofenac (Diclo). After incubating for 24 h, the supernatant (100 μL) in each well was transferred to new 96-well plates. The reagent solution containing 50 μL Griess reagent I and 50 μL Griess reagent II was added to each well, respectively. The samples on the plates were allowed to react with Griess reagent for 10 min at 27 °C in the dark. The absorbance of each sample in wells was measured at 550 nm with a Microplate Reader (Infinite M200, TECAN, Männedorf, Switzerland).

### 2.11. Measurement of the Production of Reactive Oxygen (ROS)

ROS content in the cells was determined using the ROS kit according to the manufacturer’s instructions. RAW264.7 cells were seeded in 12-well plates at a density of 1 × 10^5^/mL (1 mL per well). The cells were pre-cultured overnight at 37 °C in a 5% CO_2_ environment and then induced with 1 μg/mL LPS for 24 h. After that, the medium was replaced, and 1 mL of SP-2 or Oligo-SP-2 with concentrations of 0, 1, 5, 10, and 20 μg/mL was added to each group well, respectively, and cultured for 24 h. After removing the medium, 1 mL of a new medium containing 10 μM ROS fluorescent probe without serum was added to the cells and incubated in the dark for 45 min. After the completion of the staining, the cells were washed three times with PBS. The green fluorescence signal emitted by the ROS probe was qualitatively determined by a fluorescence inverted microscope (Olympus IX83 microscope, Tokyo, Japan).

### 2.12. Cell Migration Assay

RAW264.7 cells were inoculated into 12-well plates at a density of 1 × 10^5^ mL (1 mL per well). The cells were then observed to adhere to the bottom of each well to form a monolayer. The monolayer cells were scratched with a pipette tip, and the exfoliated cells were washed out with a PBS solution. Different concentrations of SP-2 or Oligo-SP-2 (0, 1, 5, 10, and 20 μg/mL) were added into the wells with a scratched monolayer and co-cultured for 48 h. Then the scratched monolayers were photographed, and the cell migration rate was calculated by Image J 1.51 software (NIH, Bethesda, MD, USA).
Cell migration rate (%) = (Scratch area_(0h)_) − Scratch area_(24/48 h)_) × 100%/Scratch area_(0h)_)

### 2.13. Statistical Analysis

All values were presented as the mean ± standard deviation (SD) from at least three independent experiments. Differences among groups were analyzed with one-way analysis of variance (ANOVA) and Tukey’s post-hoc multiple comparison tests. *p* < 0.05 was considered statistically significant. Values with different letters (a–f) are significantly different (*p* < 0.05).

## 3. Results and Discussion

### 3.1. The Characteristics of SP-2 after Purification

Crude polysaccharide from the body wall of *S. japonicus* was purified using a Sepharose CL-6B column. Two fractions, SP-1 and SP-2, were obtained based on their elution spectra (Figure 1A). SP-2 is the main constituent in sea cucumber polysaccharide, and the subsequent work is based on this fraction. The yield of SP-2 was 3%, and it contained 35.00% ± 0.034 carbohydrates and 32.60% ± 0.004 sulfate groups. SP-2 is composed of monosaccharides including Man, GlcN, and GalN at a ratio of 2.3:1:4.5. SP-2 was subsequently subjected to the HPGPC instrument to determine its purification and molecular weight. As shown in Figure 1B, SP-2 brought out one narrow symmetrical peak, which displayed the purification of this fraction with the MW calculated as 3275 Da. The molecular weight of Oligo-SP-2 was 2765 Da. The degree of sulfation in the Oligo-SP-2 was 3.9% ± 0.03.

### 3.2. Effect of DBD Input Voltage on SP-2

During plasma treatment, the power density of the discharge is directly affected by the pulse voltage and frequency [25]. Therefore, the input voltage levels of 60, 70, 80, and 90 V were used to study the depolymerization effect of DBD treatment on SP-2. After 5 min’ treatment by DBD at a sample concentration of 5 mg/mL with 1 A of current and various voltages, SP-2 showed two or one fraction in the HPGPC spectra. The peaks with higher retention time correspond to the smaller size of the product, and the peak area proportion was able to quantify the fraction. The increase in input voltage efficiently magnified the smallest polysaccharide peak area (Figure 2A). Improving the input voltage led to a reduction in the molecular size of the degradation product (Table 1).

The percentage of depolymerization in SP-2 after DBD treatment was calculated using the area-normalization method. The ratio of the lowest MW fraction peak area to the total peak area was used to evaluate depolymerization efficiency. The depolymerization efficiency is the area of the last peak/total area. Thus, 80% of SP-2 was converted into oligosaccharide at 90 V (Figure 2D). Due to the high voltage leading to thorough degradation, low voltage is better for depolymerization analysis. The voltage of 70 V was adopted to explore the effect of other factors on the SP-2 depolymerization.

### 3.3. Effect of DBD Treatment Time on SP-2

The effect of DBD treatment time on SP-2 depolymerization was tested for 3, 5, 7, and 9 min with an input voltage of 70 V at 1 A. The result showed that two fractions emerged on HPGPC spectra after DBD treatment (Figure 2B). The size of the depolymerization product was effectively reduced, and the proportion of the product was enlarged (Figure 2B). As calculated above, 77% of SP-2 was depolymerized into small-size products after 9 min treatment (Figure 2E). The MW distribution is shown in Table 1. The extension of the treatment time generated a smaller depolymerization product.

### 3.4. Effect of SP-2 Concentration on Depolymerization

The initial concentration of SP-2 was found to have a significant effect on the size of the depolymerization product. A series of SP-2 concentrations of 1, 3, and 5 mg/mL were used to demonstrate this effect at 70 V and 1 A for 5 min (Figure 2C). The concentration of SP-2 at 1 mg/mL exhibited the highest raw depolymerization rate, which reached 80% (Figure 2F). However, the concentration of SP-2 at 5 mg/mL yielded a smaller final product in comparison with the other two doses (Table 1). 

DBD plasma is a new physical-chemistry discharge process that can perform more homogeneous discharge [9]. The depolymerization results showed that DBD treatment generated a homogeneous polysaccharide product, which showed a very narrow peak in the HPGPC spectrum. The degradation product of other methods, such as hydrogen peroxide, showed a broad peak in the HPGPC spectrum [26]. The HPGPC spectrum shows a broad peak, indicating a mixture of multiple polymer fractions in the product. Conversely, the narrow peak of the DBD treatment product indicated the purity of the sample. The pure depolymerization product is particularly convenient for polysaccharide structure analysis. In addition, various molecular weight products were obtained by adjustment of the discharge parameters, such as the input voltage, the treatment time, and the SP-2 concentration (Table 1). This result indicated that the expected size of the polysaccharide depolymerization product can be obtained by adjusting the parameters of DBD discharge. Hence, it is a very useful tool for polysaccharide structure analysis. In addition to that, the DBD treatment of polysaccharides only requires electric power, without the need for any reagents. This results in a product not containing salts, making it easier to use it for determinations such as mass spectrometry analysis.

### 3.5. The pH Changes with the Voltage Increase and Time Extension during DBD Treatment

The pH value was measured during DBD treatment. The pH value in the SP-2 solution decreased dramatically with voltage promotion and treatment time prolongation (Figure 3). The initial pH value of the SP-2 solution was 6.83. The final pH value decreased to 1.11 when the input voltage increased to 90 V. Meanwhile, the pH dropped to 1.01 after DBD treatment for 9 min. The pH reduction of the SP-2 solution during DBD treatment is attributed to the generation of active free radicals. When the solution was subjected to DBD treatment, high-energy electron bombardment of water molecules led to active species (i.e., O_3_, O, and ·OH radicals) and ultraviolet radiation generated in the discharge plasma process [27]. The hydroxyl radicals were recombined into hydrogen peroxide, and then, in the presence of O_3_, hydrogen peroxide could be consumed and reacted to generate ·OH [28,29]. During DBD treatment, the generation and accumulation of free radicals caused a decrease in pH levels [30]. The glycosidic bond of the polysaccharide was susceptible to the active free radicals; thus, depolymerization occurred during the DBD treatment [31,32]. 

### 3.6. Reducing the End Formation of SP-2 after DBD Treatment

In this study, we used a 3,5-dinitrosalicylic acid (DNS) assay to investigate the reducing end formation of SP-2 after DBD treatment. A new reducing end will form after the polysaccharide degradation, and the amount was used to evaluate the activities of carbohydrases, such as cellulase amylases, β-mannanases, pectinases, and xyloglucanases [3]. This measurement can indicate the result of the enzymatic reaction, whether the glycosidic bond exists between the two carbohydrates or between a carbohydrate and a noncarbohydrate moiety [22]. As shown in Figure 4, a new reducing end was formed after DBD treatment of SP-2. The absorbing measurement of the reducing end increased from 0.07 to 0.11 after DBD treatment SP-2 for 7 min. The increase in the reducing end during DBD treatment indicated that the rupture site inside the SP-2 chain occurred between the two carbohydrates, at least partially. If the depolymerization of polysaccharide occurred by cutting the end sugar residues one by one, the process would experience significant weight loss. However, if the rupture site inside the SP-2 chain occurred between the two carbohydrates, the depolymerization process would retain most weight after degradation.

### 3.7. FTIR Spectra of SP-2 Depolymerization Products

The FTIR spectra of SP-2 displayed a broad stretching intensity peak at 3414.83 cm^−1^, which is attributed to the stretching vibration of O-H. The weak peak toward 2921.15 cm^−1^ was due to the C-H stretching and bending vibrations (Figure 5). These were typical absorptive bands of glycosidic structures in polysaccharide [33]. The FTIR spectra of the digested SP-2 were like the original ones, which indicated no major structural feature change after the digestion. The strengthening of the band at 1384.43 cm^−1^ in the digested SP-2 indicated the break of C-C on the side chains, which indicated the depolymerization position was inside the SP-2 chain [34].

### 3.8. Depolymerization Product Analysis by HPLC-PAD-MS

The product obtained from SP-2 after treatment for 9 min with an input voltage of 70 V at 1 A was frozen dried, derivatized by PMP, and analyzed by HPLC-PAD-MS. PMP derivatization of the sample gave six pseudo-molecular ions with *m/z* 835.42, 876.50, 997.33, 1159.58, 1181, and 996. SP-2 was composed of the monosaccharides Man and GlcN. The pseudo-molecular ions with *m/z* 835.42 and 876.50 were trisaccharides of [Man-Man-Man + H] ^+^ and [Man-Man-Man + H_2_O + Na^+^] ^+^, respectively (Figure 6A). The pseudo-molecular ions with *m/z* 997.33 were a tetrasaccharide of [Man-Man-Man-Man + H] ^+^ (Figure 6B). The pseudo-molecular ions with *m/z* 1159.58 and 1181 were pentasaccharides of [Man-Man-Man-Man-Man + H] ^+^ and [Man-Man-Man-Man-Man + H_2_O + H] ^+^, respectively (Figure 6C). The pseudo-molecular ion with *m/z* 996 was a tetrasaccharide of [GlcN-Man-Man-Man + H] ^+^ (Figure 6D). SP-2 would be digested into short-chain fragments under this extreme condition. Based on the HPLC-PAD-MS^n^ result, it would be deduced that the sequence of SP-2 has the monosaccharide sequence of GlcN-Man-Man-Man-Man-Man.

### 3.9. Digestion Results of SP-2 Using the Ultrasonic or Hydrogen Peroxide Method

No obvious molecular size change of SP-2 was observed after the treatment by using the ultrasonic or hydrogen peroxide method (Figure 7). The ultrasonic and hydrogen peroxide methods were shown to have positive degradation results in previous studies detected by using intrinsic viscosity measurement [35]. However, they were not effective on the digestion of SP-2 while the products were analyzed by HPGPC. The polysaccharide degradation results were evaluated by the measurement of intrinsic viscosity in many polysaccharide degradation studies [32,36]. The value of intrinsic viscosity was related to the number of sugar chains in the solution. The increased quantity of sugar chains indicated that the sugar residue was released during degradation. However, the slight fragments released from polysaccharide could hardly lead to an obvious molecular weight change. Hence, the intrinsic viscosity measurement may overestimate the digestion result of the polysaccharide.

### 3.10. Effect of Oligo-SP-2 on the Proliferation of RAW264.7 Cells

RAW264.7 cells play a crucial role in the host’s inflammatory and immune responses [31,37]. Oligo-SP-2 is a depolymerization product of SP-2 that has been treated by DBD. With the concentration increase of Oligo-SP-2, the cell viability showed a trend of first increase and then decrease (Figure 8). Oligo-SP-2 was less toxic to the cells. When the concentration of Oligo-SP-2 reaches 5 μg/mL, the cell viability shows the highest level of 93.98% (*p* < 0.05). However, cell viability was obviously inhibited when the concentration was greater than 20 μg/mL. Hence, a concentration within 20 μg/mL of SP-2 or Oligo-SP-2 was adopted in the subsequent cell experiments.

### 3.11. Effect of Oligo-SP-2 on the NO Release of RAW264.7 Cells

The effect of Oligo-SP-2 on regulating macrophage anti-inflammation was evaluated by measuring the accumulation of nitrite (a stable product of NO) in the supernatant of RAW264.7 macrophages. LPS-induced macrophages would secrete many pro-inflammatory cytokines to trigger an inflammatory response [38]. The NO will be secreted at the sites of inflammation by macrophages after LPS inducement, and excess NO can lead to various complications [39]. Hence, an increase in NO levels is a sign of inflammation. LPS treatment caused an improvement in NO release in the model group, which means an inflammatory response. When LPS-induced macrophages were treated with Oligo-SP-2 at concentrations of 1–20 μg/mL, the NO release was significantly alleviated compared with the LPS group (Figure 9). The NO content of the cells treated with SP-2 at a concentration of 5 μg/mL was the lowest, which decreased by 12% compared with the LPS group (*p* < 0.05). NO is an inflammatory mediator, and overproduction of NO is harmful and can cause various inflammatory and autoimmune diseases [31]. The inhibition of NO release by Oligo-SP-2 indicated that it has the function of an anti-inflammatory. The treatment of 5 μg/mL Oligo-SP-2 can effectively alleviate the occurrence of inflammation. Moreover, the results of NO showed that lower concentrations of SP-2 and Oligo-SP-2 had better anti-inflammatory ability than the SCP.

### 3.12. Effect of Oligo-SP-2 on the ROS Release Level of RAW264.7 Cells

ROS accumulation in RAW264.7 cells is a key factor in inflammatory occurrences. To verify the free radical scavenging ability of Oligo-SP-2, intracellular ROS levels in RAW264.7 cells were measured using the ROS detection probe DCFH-DA. LPS treatment led to the massive accumulation of ROS in intracellular RAW264.7 cells, which was 4.39 folds greater than the blank control group. However, the addition of Oligo-SP-2 to the LPS-induced cells intensely inhibited the accumulation of ROS. The accumulation of ROS was reduced by 3.7 folds at a concentration of 5 μg/mL compared with the LPS group (Figure 10). LPS induced intracellular ROS release in RAW264.7 cells and produced an inflammatory reaction. However, Oligo-SP-2 reduced the release of ROS in cells. Oxidative stress can activate a variety of transcription factors, which lead to the differential expression of some genes involved in inflammatory pathways [40]. ROS drive modifications of IκB proteins and allow active NF-κB to translocate to the nucleus. This action induces the expression of several molecules, which are then involved in the inflammatory process [41]. Therefore, Oligo-SP-2 alleviated the occurrence of inflammation by inhibiting the accumulation of ROS, and this activity is superior to SCP.

### 3.13. Effect of Oligo-SP-2 on the Migration of RAW264.7 Cells

To assess the Oligo-SP-2 effect on the migration of RAW264.7 cells, a cell scratch assay was performed. As shown in Figure 11, different concentrations of Oligo-SP-2 promoted cell migration to a certain extent. When the concentration was 5 μg/mL, the cell migration rates of 24 h and 48 h were the highest, reaching 16.39% and 43.34%, respectively. The migration capability of RAW264.7 cells is a sign of cell proliferation and cell vitality. The stimulation of RAW264.7 cell migration by Oligo-SP-2 indicated the anti-inflammatory force of Oligo-SP-2.

## 4. Conclusions

A DBD instrument operated under atmospheric pressure is a powerful tool for polysaccharide depolymerization. DBD treatment depolymerized SP-2 into a variety of polymers with good purity. The depolymerization ratio reached its maximum of 80%. With the help of this tool, the sequence of SP-2 was established to be GlcN-Man-Man-Man-Man-Man. The LPS-induced in vitro experiment showed that SP-2 degradation production by DBD alleviated the occurrence of inflammation by inhibiting the accumulation of ROS. In addition, the depolymerization product enhanced the migration of RAW264.7 cells, thus exerting an anti-inflammatory function.

## Figures and Tables

**Figure 1 foods-12-04079-f001:**
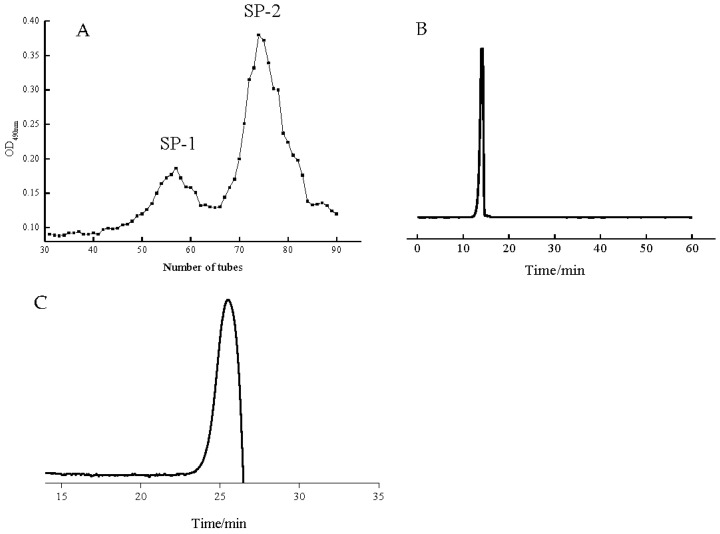
Elution spectrum of crude sea cucumber polysaccharide with the column of Sepharose CL-6B (**A**); HPGPC spectrum of SP-2 (**B**); and molecular weight of Oligo-SP-2 (**C**).

**Figure 2 foods-12-04079-f002:**
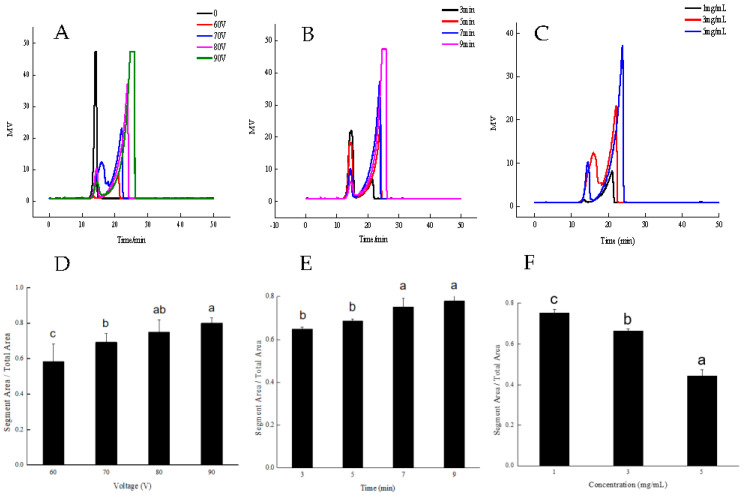
HPGPC spectra of DBD-treated SP-2 fragments. (**A**) DBD treatment of SP-2 with various voltages. (**B**) DBD treatment of SP-2 over various periods. (**C**) DBD treatment of SP-2 at various initial concentrations. (**D**–**F**) The SP-2 depolymerization percentage for each described treatment. “a–c” indicate *p* < 0.05 in different groups.

**Figure 3 foods-12-04079-f003:**
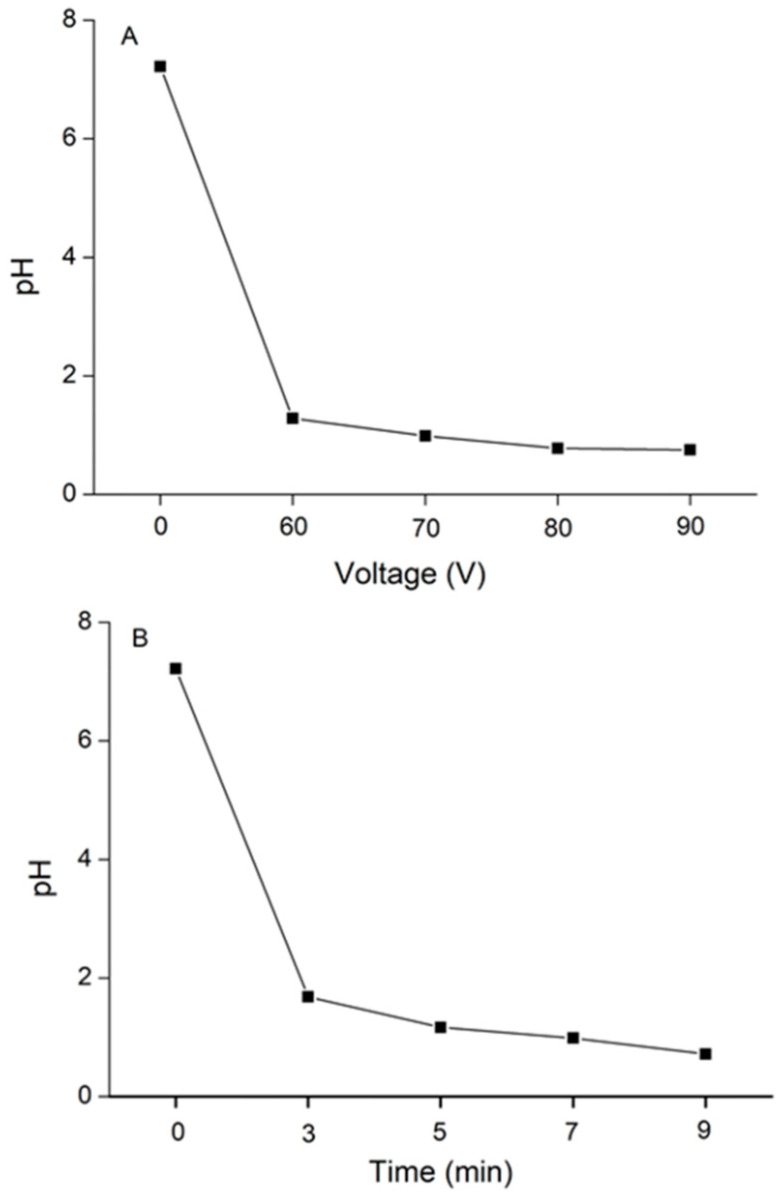
(**A**) The pH changes of SP-2 solution during DBD treatment from 0 to 90 V and (**B**) 0 to 9 min.

**Figure 4 foods-12-04079-f004:**
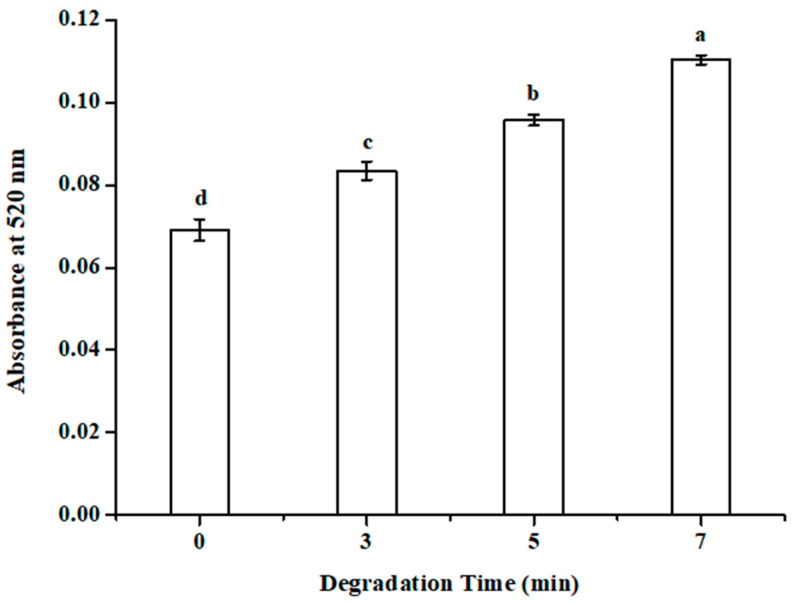
The reducing end formation in SP-2 solution after the DBD treated for 0, 3, 5, and 7 min. “a–d” means *p* < 0.05 in different groups.

**Figure 5 foods-12-04079-f005:**
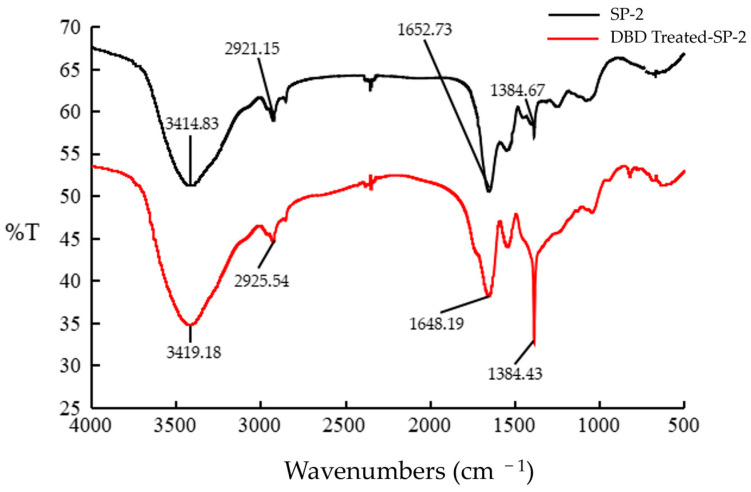
The FTIR spectra of SP-2 before and after the depolymerization with DBD.

**Figure 6 foods-12-04079-f006:**
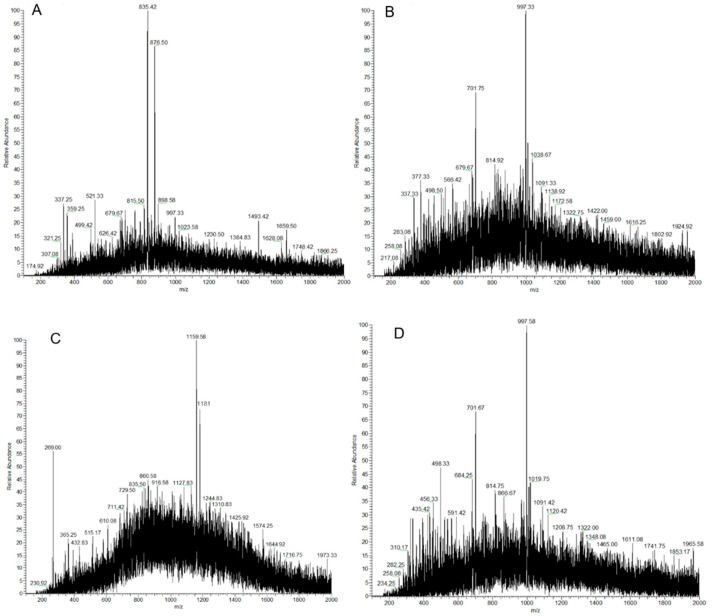
(**A**–**D**) HPLC-PAD-MS spectra of DBD-treated SP-2 for 9 min under an input voltage of 70 V at 1 A.

**Figure 7 foods-12-04079-f007:**
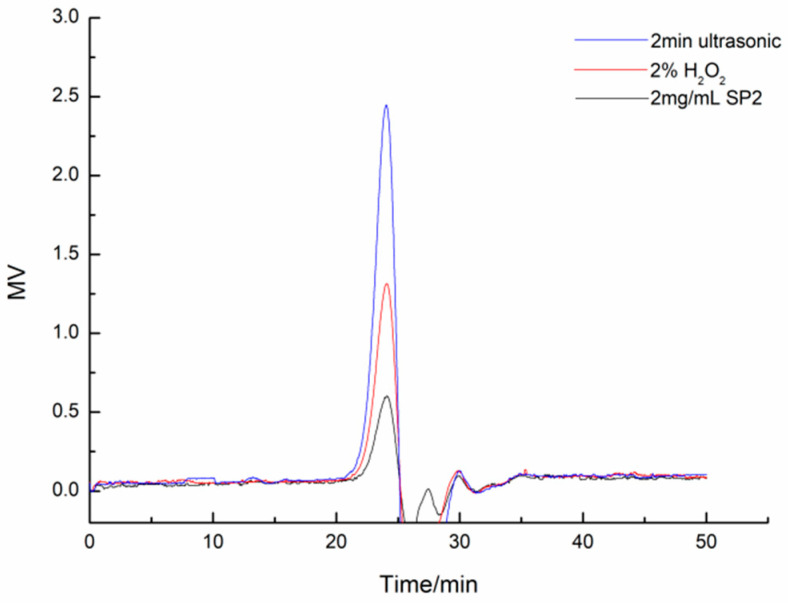
HPGPC spectra of SP-2 and digested SP-2 using the ultrasonic and hydrogen peroxide methods, respectively. SP-2 was digested by ultrasonic treatment for 20 min with a power intensity of 48.4 W/cm^2^. SP-2 was digested by hydrogen peroxide with a H_2_O_2_ concentration of 2% at 50 °C for 5 h.

**Figure 8 foods-12-04079-f008:**
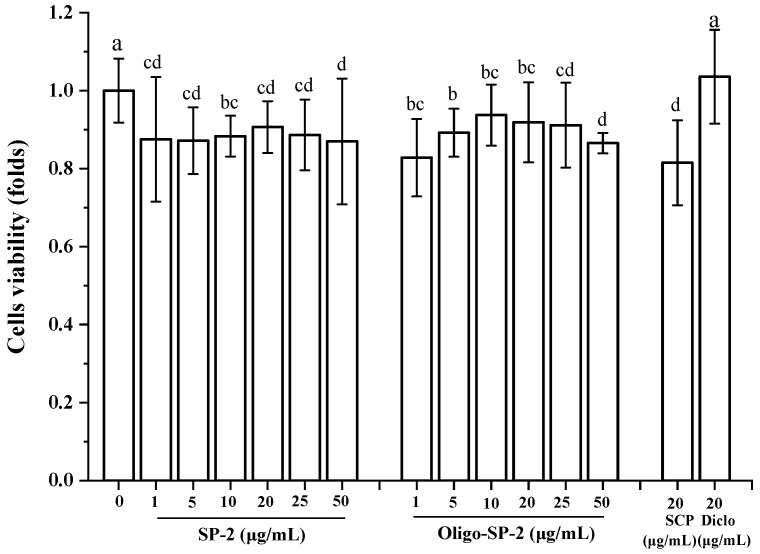
The effect of various concentrations of SP-2 or Oligo-SP-2 treatment on the viability of RAW264.7 cells. “a–d” means *p* < 0.05 in different groups.

**Figure 9 foods-12-04079-f009:**
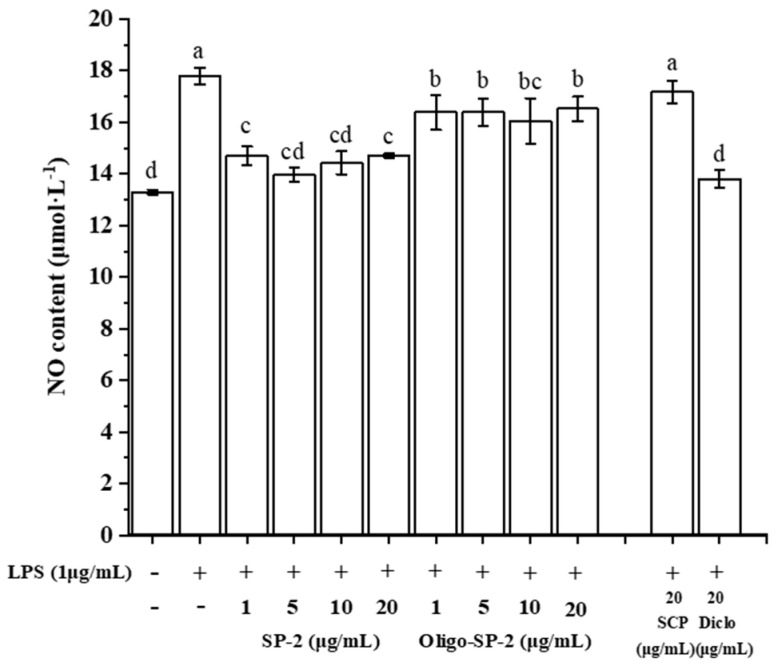
The effect of various concentrations of SP-2 or Oligo-SP-2 on NO release in RAW264.7 cells. “a–d” means *p* < 0.05 in different groups.

**Figure 10 foods-12-04079-f010:**
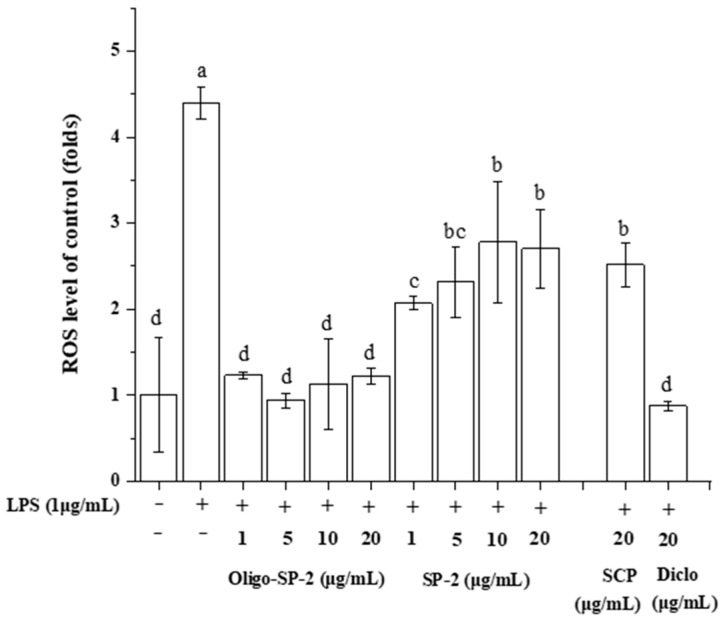
The effect of various concentrations of SP-2 or Oligo-SP-2 treatment on the ROS level of RAW264.7 cells. “a–d” means *p* < 0.05 in different groups.

**Figure 11 foods-12-04079-f011:**
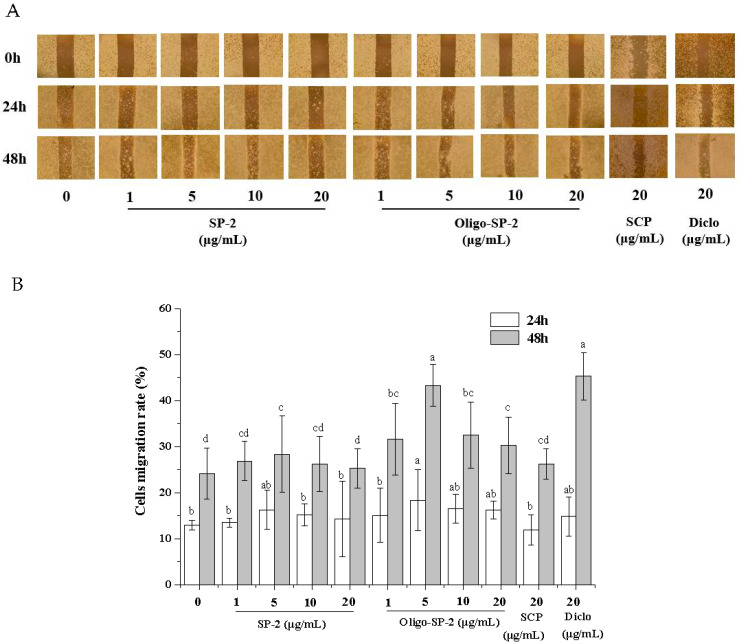
The effect of SP-2 or Oligo-SP-2 on the migration of RAW264.7 cells. (**A**) Representative images of the scratch assay (microscope magnification is 40×). (**B**) Cell migration rate (%) in different treatments. “a–d” means *p* < 0.05 in different groups.

**Table 1 foods-12-04079-t001:** Relationship of treatment time, input voltage, and sample concentration to MW of the smallest SP-2 product after treatment by DBD.

SP-2	Voltage (V) 5 min, 5 mg/mL				Time (min) 70 V, 5 mg/mL				Concentration (mg/mL) 5 min, 70 V		
	60	70	80	90	3	5	7	9	1	3	5
MW (kDa)	38.02	19.95	6.76	3.98	30.20	19.95	20.89	2.75	35.33	21.36	6.88

## Data Availability

Data is contained within the article.

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
