# Peer review of "Degradation Product of Sea Cucumber Polysaccharide by Dielectric Barrier Discharge Enhanced the Migration of Macrophage In Vitro"

_foods, 2023, doi:10.3390/foods12224079_

Round 1
Reviewer 1 Report
Comments and Suggestions for Authors
I have read the manuscript “Molecular weight-controlled degradation of sea cucumber polysaccharides by dielectric barrier discharge and anti-inflammatory properties of the products of this process” by Cheng et al., and I have several questions and suggestions.
1. The introduction must state the need for molecular weight controllable degradation of Sea Cucumber polysaccharides for food.
2. How do the authors propose to use low molecular weight polysaccharides in food?
3. How many sea cucumbers were purchased from Liujiaqiao aquatic product market?
4. What was the repetition of the extraction? Is this one experiment from 30 g?
5. Who identified Stichopus japonicus? Please include name, voucher number and where it is stored.
6. Please explain how you dissolved the sea cucumber powder (see line 79)?
7. Please provide a literary reference for the method of extraction of polysaccharides from sea cucumber? Who developed and validated it?
8. What was the yield of the purified polysaccharide?
9. How is Oligo-SP-2 different from SP-2?
10. On what basis were the polysaccharide concentrations chosen for testing? Provide literary references.
11. In Figures 8-10 there is no difference between the concentration points. Perhaps the concentration range for the study was not selected correctly.
12. The in vitro study was carried out without using the reference drug and the crude extract. Please carry out the necessary experiments to confirm your conclusions. Compare the effectiveness of your purified fraction versus the crude extract and a drug that has an anti-inflammatory effect (for example, diclofenac, etc.).
13. Please provide data confirming the possibility of using depolymerized polysaccharide from sea cucumber as food.
14. What was the purity of the depolymerized polysaccharide from sea cucumber? What were the impurities? How many impurities were there? What was the degree of sulfation?
15. The authors studied only in vitro activity. It is necessary to match the title and conclusions of the manuscript with the material presented.
16. The manuscript needs to be supplemented with additional experimental data to confirm the conclusions.
Author Response
Respond to Reviewer 1
I have read the manuscript “Molecular weight-controlled degradation of sea cucumber polysaccharides by dielectric barrier discharge and anti-inflammatory properties of the products of this process” by Cheng et al., and I have several questions and suggestions.
1.The introduction must state the need for molecular weight controllable degradation of Sea Cucumber polysaccharides for food.
We have introduced the sea cucumber polysaccharide and the need for molecular weight controllable degradation of Sea Cucumber polysaccharides for food at Lines 48-55.
2.How do the authors propose to use low molecular weight polysaccharides in food?
Sea cucumber is a high-value seafood in East Asia. This species is considered both a medicine and a food source. Low molecular weight polysaccharides can develop into a functional food. This functional food can be prepared as a capsule for usage [1].
3.How many sea cucumbers were purchased from Liujiaqiao aquatic product market?
Twelve Sea cucumbers (Stichopus japonicus, each about 180 ± 10 g) were purchased from local aquatic product market. We have added it at Lines 81-82.
4.What was the repetition of the extraction? Is this one experiment from 30 g?
The aim of the extraction is to obtain the raw material. Hence, it is unnecessary to replicate for 3 times. This is one experiment from 30 g.
5.Who identified Stichopus japonicus? Please include name, voucher number and where it is stored.
The sea cucumbers were identified as Stichopus japonicus by Professor Dayong Zhou, who works at Dalian Polytechnic University and has long experience studying sea cucumbers. The voucher number is 6937774. The sea cucumbers were freeze-dried immediately after their arrival at the lab. Then they were mashed, and stored at 4 °C for further extraction. We have added this information at Lines 82-86.
6.Please explain how you dissolved the sea cucumber powder (see line 79)?
We just put the sea cucumber powder into the 2% sodium hydroxide solution, no matter it dissolves or not. The sea cucumber powder amount was previously calculated at the 5% proportion of the total solution. We consider that it is unnecessary to add this explanation to the manuscript. But we have rewritten this sentence to make it clear at Lines 94-96.
7.Please provide a literary reference for the method of extraction of polysaccharides from sea cucumber? Who developed and validated it?
We have added a literary reference for the extraction method at Line 95. The IR experiment result validated that the extract is a polysaccharide structure.
8.What was the yield of the purified polysaccharide?
The yield of SP2 was 0.3%. we have added at Line 222.
9.How is Oligo-SP-2 different from SP-2?
Oligo-SP-2 is the depolymerization product of SP2 by the treatment of DBD. The HPGPC spectrum of Oligo-SP-2 and SP-2 was shown in Fig. 1C and Fig. 1B, which indicates that the molecular weight is reduced obviously.
10.On what basis were the polysaccharide concentrations chosen for testing? Provide literary references.
The concentration of polysaccharides used for testing is based on the proliferation of RAW264.7 cells. Fig. 8 showed that the Oligo-SP-2 was less toxic to the cells within the concentration of 20 μg/mL. Thus, the tests for subsequence were conducted in a range of concentrations from 0 to 20 μg/mL.
11.In Figures 8-10 there is no difference between the concentration points. Perhaps the concentration range for the study was not selected correctly.
The purpose of using multiple concentrations of Oligo-SP-2 in the tests is to determine their repeatability. The efficacy of Oligo-SP-2 can be assessed by comparing its results with those of the LPS group. In fact, there is a difference between the concentration points in Figures 8-10. A statistical analysis was conducted to identify any variations among the groups. Values with different letters (a-f) are significantly different (p < 0.05).
12.The in vitro study was carried out without using the reference drug and the crude extract. Please carry out the necessary experiments to confirm your conclusions. Compare the effectiveness of your purified fraction versus the crude extract and a drug that has an anti-inflammatory effect (for example, diclofenac, etc.).
We added the SP-2 group in Figures 8-11 for comparison between the purified fraction and depolymerization product. The effectiveness of the purified fraction and the depolymerization product can be evaluated by comparing it with the LPS group. We did not add a drug with an anti-inflammatory effect as a reference, since SP-2 is a food and it cannot compete with drugs. The effectiveness of the samples can be judged by comparison with the LPS group.
13.Please provide data confirming the possibility of using depolymerized polysaccharide from sea cucumber as food.
Sea cucumber is a high-value seafood in East Asia. The sea cucumber polysaccharide extract from sea cucumber is also an edible food component, which has been developed to be an oral liquid food product. Obviously, the depolymerized polysaccharide from sea cucumber also can be used as food [2,3].
14.What was the purity of the depolymerized polysaccharide from sea cucumber? What were the impurities? How many impurities were there? What was the degree of sulfation?
The depolymerized polysaccharide from sea cucumber is a well-purity fraction, which can be observed by HPGPC spectra. See Fig. 1C. The results show that the gel column chromatography elution curve for a single narrow symmetrical peak, explains homogeneous components of the product. The molecular weight of the product was estimated at 2765 Da. During the treatment of DBD, SP-2 was dissolved in distilled water without any other reagent being added. The DBD treatment is just a process of electro-discharge. Hence, no impurities are left in the product. The degree of sulfation in the product is 3.9% ± 0.03. We have added the result at Line 228-229.
15.The authors studied only in vitro activity. It is necessary to match the title and conclusions of the manuscript with the material presented.
In fact, the conclusions demonstrated that the degradation products of SP-2 exhibit anti-inflammatory properties in vitro. Thus, we have added the expression "in vitro" to the abstract and conclusion of this new manuscript. We also changed the title to avoid the misleading.
16.The manuscript needs to be supplemented with additional experimental data to confirm the conclusions.
The degradation product of SP-2 showed anti-inflammatory properties in Fig. 9-11. A lot of reports claim the anti-inflammatory properties of the extract by using in vitro experiments [4-6]. Hence, the in vitro experiment results are enough to verify this point.
[1]Li, H.; Yuan, Q.; Lv, K.; Ma, H.; Gao, C.; Liu, Y.; Zhang, S.; Zhao, L. Low-molecular-weight fucosylated glycosaminoglycan and its oligosaccharides from sea cucumber as novel anticoagulants: A review. Carbohydrate Polymers 2021, 251, 117034, doi:https://doi.org/10.1016/j.carbpol.2020.117034.
[2]Xu, C.; Zhang, R.; Wen, Z. Bioactive compounds and biological functions of sea cucumbers as potential functional foods. Journal of Functional Foods 2018, 49, 73-84, doi:https://doi.org/10.1016/j.jff.2018.08.009.
[3]Sardari, R.R.R.; Nordberg Karlsson, E. Marine Poly- and Oligosaccharides as Prebiotics. Journal of Agricultural and Food Chemistry 2018, 66, 11544-11549, doi:10.1021/acs.jafc.8b04418.
[4]Figueiredo, R.D.A.; Ortega, A.C.; GonzÁLez Maldonado, L.A.; Castro, R.D.d.; ÁVila-Campos, M.J.; Rossa Junior, C.; Aquino, S.G.d. Perillyl alcohol has antibacterial effects and reduces ROS production in macrophages. Journal of Applied Oral Science 2020, 28.
[5]Chang, S.-H.; Lin, Y.-Y.; Wu, G.-J.; Huang, C.-H.; Tsai, G.J. Effect of chitosan molecular weight on anti-inflammatory activity in the RAW 264.7 macrophage model. International Journal of Biological Macromolecules 2019, 131, 167-175, doi:https://doi.org/10.1016/j.ijbiomac.2019.02.066.
[6]Wang, Y.; Tian, Y.; Shao, J.; Shu, X.; Jia, J.; Ren, X.; Guan, Y. Macrophage immunomodulatory activity of the polysaccharide isolated from Collybia radicata mushroom. International Journal of Biological Macromolecules 2018, 108, 300-306, doi:https://doi.org/10.1016/j.ijbiomac.2017.12.025.
Reviewer 2 Report
Comments and Suggestions for Authors
General comments
The title of the manuscript seems too long and not attractive
The abstract did not include any description for treatment conditions of DBD
The results were not discussed. There was just a description for the results
Why authors selected 70 V DBD treatment to study the effect of time and the effect of different concentrations SP2 on depolymerization pattern
Specific comments
Line 28-29: The phrase is vague. It should be restructured
Line 31-35: The phrase is very long and vague. It should be restructured and fragmentation is need
Line 40: “60Co irradiation” should be corrected to Cobalt-60 (60Co)
Line 44-45: Chemical Engineering Journal 44 317 (2017) 90–102), what do you mean with these numbers
Line 48-57: what do the authors mean with this paragraph. I did not understand its role. It seems narrative
Line 69: “abovel chemicals”, I think there is a spelling error here. Do you mean “above”, if yes, it should be below
Line 88: incubated mixture sat for 12 h at 4°C, it is better to write the mixture is hold for …., or try to write in different way.
Line 93: according to Sevag, what do you mean with Sevag If reference, please cite it.
Line 162: in100μL D, leave space
Line 205-206: The phrase is vague and incomplete in the meaning. It should be restructured
Figure 2 D-F, superscript letters should be introduced to differentiate if there were significant differences between different processing conditions
In figure 11 B: there is no legends
Comments on the Quality of English Language
The quality of English language should be improved. There is a difficulty in readability of some phrases.
Author Response
Respond to Reviewer 2
1.The title of the manuscript seems too long and not attractive
We have changed the title into Degradation Product of Sea Cucumber Polysaccharide by Dielectric Barrier Discharge Enhanced the Migration of Macrophage.
2.The abstract did not include any description for treatment conditions of DBD
Thank you for your reminder. In order to determine the optimal conditions for DBD degradation, experiments were performed for a range of voltages and degradation times. We have added the treatment conditions (60~90 V, 3~9 min) at Lines 126-127.
3.The results were not discussed. There was just a description for the results
Thank you for your sincerely check. The relevant discussion has been re-added in the results (at Lines 219-438), and the details can be found in the newly uploaded manuscript.
4.Why authors selected 70 V DBD treatment to study the effect of time and the effect of different concentrations SP2 on depolymerization pattern
Due to the high voltage led to thorough degradation, low voltage better for depolymerization analysis. The voltage of 70 V was adopted to explore the other factors effect on the SP-2 depolymerization. We have added the explanation at Lines 247-249.
Specific comments
1.Line 28-29: The phrase is vague. It should be restructured
We have restructured this phrase as you suggested at Lines 32-35.
2.Line 31-35: The phrase is very long and vague. It should be restructured and fragmentation is need
We have restructured and condensed the phrase as you suggested at Lines 38-43.
3.Line 40: “60Co irradiation” should be corrected to Cobalt-60 (60Co)
We have corrected this noun at Line 40.
4.Line 44-45: Chemical Engineering Journal 44 317 (2017) 90–102), what do you mean with these numbers
This number is a reference. We have revised this error at Line 45.
5.Line 48-57: what do the authors mean with this paragraph. I did not understand its role. It seems narrative
Thank you for your suggestion. We have made appropriate adjustments the paragraph according to your suggestion at Lines 48-55.
6.Line 69: “abovel chemicals”, I think there is a spelling error here. Do you mean “above”, if yes, it should be below
Thank you for your kind reminder. We have changed this wrong word to “above” at Line 87.
7.Line 88: incubated mixture sat for 12 h at 4°C, it is better to write the mixture is hold for …., or try to write in different way.
We have altered this sentence legitimately according to your suggestion at Line 104.
8.Line 93: according to Sevag, what do you mean with Sevag If reference, please cite it.
Sevag is a method to remove the dissociative protein. We have cited appropriate references based on your comments at Line 109.
9.Line 162: in100μL D, leave space
We have leave space according to your suggestion at Line 179. We also revised this error throughout the manuscript.
10.Line 205-206: The phrase is vague and incomplete in the meaning. It should be restructured
We have restructured this sentence at Lines 219-222.
11.Figure 2 D-F, superscript letters should be introduced to differentiate if there were significant differences between different processing conditions
Figure 2 D-F are the calculation results based on Figure 2 A-C, respectively. Thus, it cannot be statistical analyzed because of the absent of replications. These calculation results can evaluate the degradation proportion, and it is meaningful to exhibit it.
12.In figure 11 B: there is no legends
The figure legends were added at Lines 432-435.
Reviewer 3 Report
Comments and Suggestions for Authors
Introduction
Page 1, lines 38-40 – Please revise this sentence for clarification.
Page 2, lines 43-47 – Please clarify this paragraph.
Page 2, lines 65 and 66 – Please improve the end of sentence.
Materials and Methods
Page 2, lines 69 and 79 – You have not to mention “researchers”.
Page 2, line 75 – Please delete “this study”.
Page 2, line 79 – I suppose that sea cucumber powder is not water soluble but it forms a suspension in water. Please check.
Page 2, lines 80 and 81 – I think the pH of the solution was adjusted.
Page 2, line 82 – I suggest deleting “to retain the supernatant”.
Page 2, line 91 – I suppose it is “...supernatant was freeze dried” and I suggest “obtain” instead of “yield”.
Page 3, line 93 – Please clarify this sentence.
Page 3, line 94 – I suppose it is: “...sample was purified by gel...”
Page 3, line 97 – I also suggest “obtain” and “fraction” instead of “yield” and “fractions”, respectively.
Page 3, lines 108-111 – Please clarify the sentence “SP2 was... investigated”. Please indicate the volume of water solutions used in the assays.
Page 3, lines 125 and 126 – Please replace “was” with “were”.
Page 3, lines 133-137 –Please clarify these sentences.
Page 4, lines 138 and 141 – It is “distilled”.
Page 4, line 140 – I suggest including “at 520 nm” after “...Switzerland)”.
Page 4, lines 148 and 149 – I suggest deleting “was” after “SP2” and replacing “to obtain the product of” with “was designated as”. Please check.
Page 4, lines 162-173 – Please improve these sentences for clarification.
Results
Page 5, lines 205 and 206 – This sentence doesn’t seem relevant.
Page 5, line 208 – Why was the SP2 fraction selected for the further trials? And why “ingredient”?
Page 6, lines 219 and 220 – The sentence “The reference... treatment” is not clear. Please improve it.
Page 6, line 227 – Is it “The increase of input voltage efficiently...”?
Page 6, line 232 – Please indicate the concentration of the samples used in these trials.
Page 7, Table 1 – Please include in the first line: the time and concentration below “Voltage”; the voltage and concentration below “Time”; and the voltage and time bellow “Concentration”.
Page 7, line 261 – Is it “voltage increase”? Please check.
Page 7, line 262 – I suggest “was 6.83”.
Page 7, lines 267-271 – The increase of H+ concentration is not explained.
Page 8, line 277 – Is it “had formed after...”? Please check.
Page 8, line 282 – Is it “...end was formed...”? Please check.
Page 8, line 284 – I suggest “increase” instead of “growth”.
Page 9, 292 – I also suggest “...which is attributed to...”
Page 9, line 296 – I think it is “strengthening”. Please check.
Page 10, line 302 – Please consider this alternative: “The product obtained from SP2 after treatment for 9 min...1 A was frozen dried... HPLC-PAD-MS”.
Page 10, lines 308 and 311 – I suppose it is “...was a tetrasaccharide of...”
Page 10, lines 312 and 313 – The sentence “The thoroughly... 1 A” seems irrelevant.
Page 10, line 314 – I suggest “conditions” instead of “situation” and “it would be deduced” instead of “we can deduce”.
Page 10, line 324 – Please consider this alternative: “...were analyzed by HPGPC”.
Page 11, line 337 – The preparation of oligomer Oligo-SP-2 is not mentioned in the manuscript. It is just introduced in page 4, line 149.
Page 11, line 344 – I think it is “subsequent”.
Page 12, lines 368-380 – Please improve this discussion avoiding repetitions.
Comments on the Quality of English LanguageThe English language should be improved.
Author Response
Respond to Reviewer 3
Introduction
1.Page 1, lines 38-40 – Please revise this sentence for clarification.
We have revised this sentence at Lines 38-43.
2.Page 2, lines 43-47 – Please clarify this paragraph.
We have modified the paragraph at Lines 44-47.
3.Page 2, lines 65 and 66 – Please improve the end of sentence.
We have optimized the ending sentence to make the sentence more understandable at Lines 75-78.
Materials and Methods
1.Page 2, lines 69 and 79 – You have not to mention “researchers”.
We have removed the word of “researchers” and rewritten this paragraph at Lines 81 and 96.
2.Page 2, line 75 – Please delete “this study”.
We have deleted “this study” as your suggestion at Line 90.
3.Page 2, line 79 – I suppose that sea cucumber powder is not water soluble but it forms a suspension in water. Please check.
You are correct. The sea cucumber powder cannot completely dissolve in sodium hydroxide. We have replaced the word of dissolve to add at Lines 94-96.
4.Page 2, lines 80 and 81 – I think the pH of the solution was adjusted.
According to previous reference [1], the pH of the solution was adjusted to 7 at Line 97.
5.Page 2, line 82 – I suggest deleting “to retain the supernatant”.
We have deleted the phrase according to your suggestion.
6.Page 2, line 91 – I suppose it is “...supernatant was freeze dried” and I suggest “obtain” instead of “yield”.
We have altered the phase according to your suggestion at Line 106-107.
7.Page 3, line 93 – Please clarify this sentence.
We have revised it to make the sentence clear at Lines 108-109.
8.Page 3, line 94 – I suppose it is: “...sample was purified by gel...”
We have revised this sentence according to your suggestion at Line 110.
9.Page 3, line 97 – I also suggest “obtain” and “fraction” instead of “yield” and “fractions”, respectively.
We have modified the words according to your suggestion at Line 113.
10.Page 3, lines 108-111 – Please clarify the sentence “SP2 was... investigated”. Please indicate the volume of water solutions used in the assays.
The volume of water was 1 mL, we have added it and revised the sentences according to your suggestion at Line 125.
11.Page 3, lines 125 and 126 – Please replace “was” with “were”.
We have replaced “was” with “were” at Line 143.
12.Page 3, lines 133-137 –Please clarify these sentences.
We have modified the sentences at Lines 149-151.
13.Page 4, lines 138 and 141 – It is “distilled”.
We have modified the words according to your suggestion at Line 156.
14.Page 4, line 140 – I suggest including “at 520 nm” after “...Switzerland)”.
We have restructured the sentence according to your suggestion at Lines 155 and 157.
15.Page 4, lines 148 and 149 – I suggest deleting “was” after “SP2” and replacing “to obtain the product of” with “was designated as”. Please check.
We have modified the sentence according to your suggestion at Lines 166.
16.Page 4, lines 162-173 – Please improve these sentences for clarification.
We have modified the sentence according to your suggestion at Linse 176-186.
Results
1.Page 5, lines 205 and 206 – This sentence doesn’t seem relevant.
We have changed the title to “The characteristic of SP-2 after Purification” according to your suggestion at Line 218.
2.Page 5, line 208 – Why was the SP2 fraction selected for the further trials? And why “ingredient”?
SP-2 is the main constituent in sea cucumber polysaccharide, and the subsequent work is based on this fraction. We have added the sentences according to your suggestion at Lines 221-222. We have revised the sentence and removed the word of “ingredient”.
3.Page 6, lines 219 and 220 – The sentence “The reference... treatment” is not clear. Please improve it.
We modified the sentence appropriately according to your suggestion at Line 234.
4.Page 6, line 227 – Is it “The increase of input voltage efficiently...”?
We have altered the correct word according to your suggestion at Lines 240-241.
5.Page 6, line 232 – Please indicate the concentration of the samples used in these trials.
The concentration of the samples used in the trials was 5 mg/mL at Line 237.
6.Page 7, Table 1 – Please include in the first line: the time and concentration below “Voltage”; the voltage and concentration below “Time”; and the voltage and time bellow “Concentration”.
Thank you for your suggestion, and we have adjusted the contents of Table 1 according to your suggestion.
7.Page 7, line 261 – Is it “voltage increase”? Please check.
We have changed the sentence at Line 294.
8.Page 7, line 262 – I suggest “was 6.83”.
We have modified the words at Line 293.
9.Page 7, lines 267-271 – The increase of H+ concentration is not explained.
During DBD treatment, the generation and accumulation of free radicals caused a de-crease in pH levels. We have added content at Lines 301-302.
10.Page 8, line 277 – Is it “had formed after...”? Please check.
We consider that the sentence at Line 309 is a subtitle, Hence, it is inappropriate to appear the word of “had formed after...” It is better to maintain the present status.
11.Page 8, line 282 – Is it “...end was formed...”? Please check.
We have modified the sentence according to your suggestion at Line 314.
12.Page 8, line 284 – I suggest “increase” instead of “growth”.
We have altered the word according to your suggestion at Line 316.
13.Page 9, 292 – I also suggest “...which is attributed to...”
We have added “is” according to your suggestion at Line 327.
14.Page 9, line 296 – I think it is “strengthening”. Please check.
We have modified the word according to your suggestion at Line 331.
15.Page 10, line 302 – Please consider this alternative: “The product obtained from SP2 after treatment for 9 min...1 A was frozen dried... HPLC-PAD-MS”.
We have modified the sentences according to your suggestion at Lines 337-338.
16.Page 10, lines 308 and 311 – I suppose it is “...was a tetrasaccharide of...”
We have added “a” at the places you suggested at Lines 343 and 346.
17.Page 10, lines 312 and 313 – The sentence “The thoroughly... 1 A” seems irrelevant.
We have deleted this sentence as you suggested.
18.Page 10, line 314 – I suggest “conditions” instead of “situation” and “it would be deduced” instead of “we can deduce”.
We have modified the sentence according to your suggestion at Lines 348-349.
19.Page 10, line 324 – Please consider this alternative: “...were analyzed by HPGPC”.
We have altered the word according to your suggestion at Line 358.
20.Page 11, line 337 – The preparation of oligomer Oligo-SP-2 is not mentioned in the manuscript. It is just introduced in page 4, line 149.
To prepare Oligo-SP-2, follow the same procedure as described in section 2.3. We have described the treatment parameters at Lines 124-129.
21.Page 11, line 344 – I think it is “subsequent”.
We have modified the word according to your suggestion at Line 379.
22.Page 12, lines 368-380 – Please improve this discussion avoiding repetitions.
We have removed the redundant discussion. Please see the new manuscript.
[1] Wang, Y.; Sun, J.; Zhang, Y.; Liu, W.; Yang, S.; Yang, J. Stichopus japonicus Polysaccharide Stimulates Osteoblast Differentiation through Activation of the Bone Morphogenetic Protein Pathway in MC3T3-E1 Cells. Journal of Agricultural and Food Chemistry 2021, 69, 2576-2584, doi:10.1021/acs.jafc.0c06466.
Round 2
Reviewer 1 Report
Comments and Suggestions for Authors
I have read the revision manuscript. The authors did not fully answer a number of my questions.
1. The introduction must state the need for molecular weight controllable degradation of Sea Cucumber polysaccharides for food. From the introduction and the data presented, it is not clear why sea cucumber should be hydrolyzed with a yield of 0.3%. What else? Is this a waste?
2. Reproducibility is an integral part of scientific research. Reproducibility should be tested and data reported.
3. In Figures 8-10 there is no difference between the concentration points. Perhaps the concentration range for the study was not selected correctly. Please provide the primary statistical data.
4.The in vitro study was carried out without using the reference drug and the crude extract. Please carry out the necessary experiments to confirm your conclusions. Compare the effectiveness of your purified fraction versus the crude extract and a drug that has an anti-inflammatory effect (for example, diclofenac, etc. or suitable food supplement).
5. Please provide data confirming the possibility of using depolymerized polysaccharide from sea cucumber as food. What is the toxicity of your product?
Author Response
Dear Editor and Reviewer,
We appreciate for this chance to revise our manuscript to Foods and thank again to the reviewers’ suggestions. We have very carefully read the comments and made necessary corrections according to their suggestions. The main corrections in this paper and the responds to the reviewers’ comments are as flowing:
Respond to Reviewer 1
- The introduction must state the need for molecular weight controllable degradation of Sea Cucumber polysaccharides for food. From the introduction and the data presented, it is not clear why sea cucumber should be hydrolyzed with a yield of 0.3%. What else? Is this a waste?
Thank you very much for your careful review. We have stated the need for molecular weight controllable degradation of sea cucumber polysaccharides for food at Lines 36-37. The molecular weight controllable degradation of Sea Cucumber polysaccharides helps for the molecular structure analysis. The degradation product of Sea Cucumber polysaccharides provides stronger bioactivity and facility for absorption getting through intestinal mucosa barrier. The sea cucumber has long history been considered as both medicine and food in East Asia. The hydrolysate of sea cucumber polysaccharide can bring more stronger bioactivities. Hence, the low yield result can be accepted under this situation. A lot of studies have reported the similar result. The physiological activity of degraded small molecular weight polysaccharides has been reported in many aspects such as anti-obesity and antioxidant [1,2]. Although the yield of purified or degraded polysaccharide was lower than that of crude polysaccharide, its absorptivity and activity could be improved. Thus, this is not a waste, but another way to use its features more efficiently.
Besides, the yield was 3% but not 0.3%. We are sorry for this error. fresh sea cucumbers were freeze-dried and 30 g dry powder was collected. Then, 3 g crude polysaccharide of sea cucumbers was extracted by “method 2.2.”, and the yield of crude polysaccharide was 10%. Furthermore, two main components (SP-1 and SP-2) were obtained by CL-6B column separation and purification. Among them, 100 mg of crude polysaccharide was purified by CL-6B column, dialysis and lyophilized to obtain 3 mg of SP-2. Therefore, SP-2 accounted for 0.3% {100% *[3 mg/(100 mg/10%)] = 0.3%} of the initial sea cucumber dry powder yield and 3% [100% *(3 mg/100 mg) = 3%] of sea cucumber crude polysaccharide yield. We are very sorry that we did not specify the specific details of SP-2 purification yield in the “results 3.1.”, and thank you again for your correction. We have made corrections in the manuscript at Line 225.
- Reproducibility is an integral part of scientific research. Reproducibility should be tested and data reported.
All bioactivity data in this study are tested for 3 replicates. This work coincided the prevailing standard of statistical analysis. Therefore, our study has good reproducibility.
- In Figures 8-10 there is no difference between the concentration points. Perhaps the concentration range for the study was not selected correctly. Please provide the primary statistical data.
We have provided the primary statistics in Fig 8-10, please see the figure below. The results of cell experiments showed that lower concentrations of SP-2 and Oligo-SP-2 had better anti-inflammatory ability.
The values were shown as means ± standard deviation of triplicate replicates. One-way ANOVA and Tukey’s test were adopted to measure significant differences by using SPSS 22.0 software. And a-d means differences between groups (p < 0.05 was considered significant).
- The in vitro study was carried out without using the reference drug and the crude extract. Please carry out the necessary experiments to confirm your conclusions. Compare the effectiveness of your purified fraction versus the crude extract and a drug that has an anti-inflammatory effect (for example, diclofenac, etc. or suitable food supplement).
Thank you very much for your advice. We have supplemented the experiment in the RAW 264.7 cells of diclofenac (Diclo) and crude polysaccharide (SCP) of sea cucumber in this edit-manuscript.
- Please provide data confirming the possibility of using depolymerized polysaccharide from sea cucumber as food. What is the toxicity of your product?
The polysaccharide from sea cucumber has been developed as a commercial beverage food product in China (See image below). Since the polysaccharide component food is an emerging industry, we have not found the depolymerized polysaccharide from sea cucumber as food. However, many depolymerized polysaccharide products as food are selling in the market, for example, the chitosan oligosaccharide, the xylo-oligosaccharide, and the soybean oligosaccharides. In addition, a great many studies have reported the possibility of using depolymerized polysaccharides as food [3-5]. Gentiooligosaccharides (GnOS) were synthesized by the acceptor reaction of dextransucrase from Leuconostoc mesenteroides NRRL B-1426 with gentiobiose and sucrose [6], which can be used as a supplement for functional foods with anti-cancer properties.
Theoretically speaking, the depolymerization product does not have toxicity. The depolymerization process costs only electric power without any chemical reagent added. The slight toxicity in MTT experiment should be the residual free radical after degradation. This residual free radical can be removed by dialysis.
[1] Teng, C., Shi, Z., Yao, Y., Ren, G. Structural characterization of quinoa polysaccharide and its inhibitory effects on 3T3-L1 adipocyte differentiation. Foods 2020, 9, 1511, doi:10.3390/foods9101511.
[2] Yan, S., Pan, C., Yang, X., Chen, S., Qi, B., Huang, H. Degradation of codium cylindricum polysaccharides by H2O2-Vc-ultrasonic and H2O2-Fe2+-ultrasonic treatment: structural characterization and antioxidant activity. International Journal of Biological Macromolecules 2021, 182, 129-135, doi: https://doi.org/10.1016/j.ijbiomac.2021.03.193.
[3] J, Remón., Li, T., Chuck, C., Matharu, A., Clark, J. Toward renewable-based, food-applicable prebiotics from biomass: a one-step, additive-free, microwave-assisted hydrothermal process for the production of high purity xylo-oligosaccharides from beech wood hemicellulose. ACS Sustainable Chemistry & Engineering 2019, 7, 16160–16172, doi: https://doi.org/10.1021/acssuschemeng.9b03096.
[4] Pang, B., Wang, H., Huang, H., Liao, L., Wang, Y., Wang, M., Du, G., Kang. Z. Enzymatic production of low-molecular-weight hyaluronan and its oligosaccharides: a review and prospects. Journal of Agricultural and Food Chemistry 2022, 70, 14129–14139, doi: https://doi.org/10.1021/acs.jafc.2c05709.
[5] Hu, H., Zhang, S., Pan, S. Characterization of citrus pectin oligosaccharides and their microbial metabolites as modulators of immunometabolism on macrophages. Journal of Agricultural and Food Chemistry 2021, 69, 8403–8414, doi: https://doi.org/10.1021/acs.jafc.1c01445.
[6] Kothari, D., Goyal, A. Gentio-oligosaccharides from Leuconostoc mesenteroides NRRL B-1426 dextransucrase as prebiotics and as a supplement for functional foods with anti-cancer properties. Food Function 2015, 6, 604-611, doi: https://doi.org/10.1039/C4FO00802B.

Reviewer 3 Report
Comments and Suggestions for Authors
Abstract
Page 1, line 24 – I suggest “…technique can be used for...”
Introduction
Page 2, lines 52 and 53 – I think it is “…polysaccharides through...”
Page 2, line 55 – I suggest deleting “benefits”.
Page 2, line 60 – I also suggest “...to be participating in the first...”
Page 2, line 78 – I think it is “... by the degraded product”.
Materials and Methods
Page 3, line 95 – Please consider this alternative: “Thirty g of sea cucumber powder were added into 2 %...”
Page 3, line 108 – I suppose it is “dissociate” or “hydrolysed proteins”. Please check.
Page 3, lines 124 and 125 – I think the sentence “”SP-2 was dissolved...” was better in the version 1. I also suggest “One mL of the solution...”
Page 4, lines 151 and 152 – Please consider this alternative sentence: “One mL of these sample solutions...”
Page 4, lines 164 and 165 – Please also take into consideration this alternative: “To prepare the depolymerisation product for this assay, the same procedure described in section 2.3 was followed.”
Page 4, lines 179 and 180 – Please consider this alternative sentence: “After this incubation period, The culture medium was replaced with 100 µL of SP-2, Oligo-SP-2 and LPS solution and incubated for 24 h.”
Page 6, line 249 – Please consider replacing “the other factors effect” with “the effect of other factors”.
Page 6, line 256 – It is “MW”.
Page 7, Table 1, 2nd line and 3rd column – It is shown “60 V”, but in line 252 is mentioned “70 V”. Please check.
Page 7, Table 1, 2nd line and 1st column – It is “MW”.
Page 7, lines 279 and 280 – I suggest replacing “Be different from that” with “Conversely”.
Page 8, line 283 – I suggest replacing “parameter” with “parameters”.
Page 8, line 284 – Please consider replacing “...the researchers can obtain...” with “...it can be obtained...”
Page 8, line 287 – It is “...that, the DRD...”
Page 8, line 288 and 289 – Please consider this alternative: “...in a product not containing salts making easier to use it for determinations...”
Comments on the Quality of English LanguageThe quality of English language was considerably improved in the second version of the manuscript, bu there still a few points to be improved.
Author Response
Respond to Reviewer 3
Abstract
Page 1, line 24 – I suggest “…technique can be used for...”
Thank you for your suggestion, we changed the sentence at Line 24.
Introduction
Page 2, lines 52 and 53 – I think it is “…polysaccharides through...”
We have altered this sentence according to your reminder at Line 53.
Page 2, line 55 – I suggest deleting “benefits”.
We have deleted this word at Line 56.
Page 2, line 60 – I also suggest “...to be participating in the first...”
We have changed this sentence according to your suggestion at Line 61.
Page 2, line 78 – I think it is “... by the degraded product”.
We have changed this word to “degraded” at Line 78.
Materials and Methods
Page 3, line 95 – Please consider this alternative: “Thirty g of sea cucumber powder were added into 2 %...”
Thank you for your advice and this sentence has been altered at Line 96.
Page 3, line 108 – I suppose it is “dissociate” or “hydrolysed proteins”. Please check.
We have changed the word to “hydrolyzed proteins” at Line 109.
Page 3, lines 124 and 125 – I think the sentence “SP-2 was dissolved...” was better in the version 1. I also suggest “One mL of the solution...”
We have changed to “One mL” at Line 126.
Page 4, lines 151 and 152 – Please consider this alternative sentence: “One mL of these sample solutions...”
Thank you for your advice and this sentence has been reversed at Lines 152-153.
Page 4, lines 164 and 165 – Please also take into consideration this alternative: “To prepare the depolymerisation product for this assay, the same procedure described in section 2.3 was followed.”
We have changed this sentence at Lines 165-166.
Page 4, lines 179 and 180 – Please consider this alternative sentence: “After this incubation period, The culture medium was replaced with 100 µL of SP-2, Oligo-SP-2 and LPS solution and incubated for 24 h.”
This sentence has been changed at Lines 181-182.
Page 6, line 249 – Please consider replacing “the other factors effect” with “the effect of other factors”.
It has been replaced at Line 251.
Page 6, line 256 – It is “MW”.
Thank you for your careful reminder, we have corrected this error at Line 258.
Page 7, Table 1, 2nd line and 3rd column – It is shown “60 V”, but in line 252 is mentioned “70 V”. Please check.
We are very sorry and have corrected this error in Table 1.
Page 7, Table 1, 2nd line and 1st column – It is “MW”.
We changed “Mw” to “MW” in Table 1.
Page 7, lines 279 and 280 – I suggest replacing “Be different from that” with “Conversely”.
This word has been changed at Line 281.
Page 8, line 283 – I suggest replacing “parameter” with “parameters”.
We have changed this word at Line 285.
Page 8, line 284 – Please consider replacing “...the researchers can obtain...” with “...it can be obtained...”
This sentence has been alerted at Line 286.
Page 8, line 287 – It is “...that, the DRD...”
Thank you for your reminder. We corrected this error at Line 288.
Page 8, line 288 and 289 – Please consider this alternative: “...in a product not containing salts making easier to use it for determinations...”
We have changed this sentence at Line 290.